# Design and Implementation of a Prototype Seismogeodetic System for Tectonic Monitoring [note 1]

**DOI:** 10.3390/s23218986

**Published:** 2023-11-05

**Authors:** Javier Ramírez-Zelaya, Belén Rosado, Vanessa Jiménez, Jorge Gárate, Luis Miguel Peci, Amós de Gil, Alejandro Pérez-Peña, Manuel Berrocoso

**Affiliations:** 1Laboratory of Astronomy, Geodesy and Cartography, Department of Mathematics, Faculty of Sciences, University of Cadiz, 11510 Puerto Real, Spain; belen.rosado@uca.es (B.R.); jorge.garate@uca.es (J.G.); luismiguel.peci@uca.es (L.M.P.); amos.degil@uca.es (A.d.G.); alejandro.perezpena@uca.es (A.P.-P.); manuel.berrocoso@uca.es (M.B.); 2Department of Theoretical Physics and the Cosmos, University of Granada, 18010 Granada, Spain; vanessa.jimenezmorales@hotmail.com

**Keywords:** seismogeodetic systems, time series analysis, tectonic surveillance, earth deformation, seismic hazard, tsunami hazard

## Abstract

This manuscript describes the design, development, and implementation of a prototype system based on seismogeodetic techniques, consisting of a low-cost MEMS seismometer/accelerometer, a biaxial inclinometer, a multi-frequency GNSS receiver, and a meteorological sensor, installed at the Doñana Biological Station (Huelva, Spain) that transmits multiparameter data in real and/or deferred time to the control center at the University of Cadiz. The main objective of this system is to know, detect, and monitor the tectonic activity in the Gulf of Cadiz region and adjacent areas in which important seismic events occur produced by the interaction of the Eurasian and African plates, in addition to the ability to integrate into a regional early warning system (EWS) to minimize the consequences of dangerous geological phenomena.

## 1. Introduction

The seismic records generated by accelerometers, short-period, and broadband seismometers provide us with real-time location, depth, magnitude, epicenter, etc., of a seismic event and the possibility of assessing the occurrence of tsunamis. Certain limitations have been observed in many of these sensors in high-magnitude earthquakes, the most common being the possibility of saturation in the frequency band of the seismic signal, generating uncertainty in the results obtained; in terms of the resolution of the accuracy of the co-seismic displacement, however, the Global Navigation Satellite System (GNSS) has the ability to measure land surface displacement using higher frequency bands, providing support to seismic networks in determining the velocity and displacement of the occurred event [1,2,3].

The high sampling frequencies of current GNSS sensors (up to 10 Hz) make GNSS observations available to directly measure the displacements caused by seismic activity. These high-frequency observations show that the GNSS-GPS system is an excellent tool for measuring large displacements in areas near earthquakes, where the seismographs due to the limits in their dynamic range are saturated, impeding correct calculation of location and magnitude, when, in fact, this information is basic for the detection and rapid evaluation of the seismic event. While both fields have made substantial progress independently, the integration of seismology and geodesy has led to the development of seismogeodetic systems, a revolutionary approach that offers a more holistic understanding of seismic events and their impact. The seismogeodetic systems based on the integration of GNSS receivers and accelerometers complement seismic networks in moderate-magnitude earthquakes but will be essential to the occurrence of high-magnitude earthquakes [4].

Seismogeodetic systems combine the principles and methodologies of seismology and geodesy to capture the complete situation of an earthquake. They incorporate a diverse range of technologies, including seismometers and GNSS Systems, to observe and analyze ground motion, deformation, and seismic events. These systems are essential for understanding and mitigating the impacts of seismic and tectonic events. Moreover, seismogeodetic systems offer essential data for understanding the dynamics of tectonic activity, including the accumulation of strain along fault lines and the rate of ground deformation.

Early warning systems (EWS) include knowledge of risk, monitoring and alerts, diffusion, and response capacity; those most directly linked to the processes monitoring and surveillance of tectonic activity are knowledge of risk and monitoring together with warning generation [5]. A priority of the EWS should be the ability to provide immediate information on the tectonic activity based on the deformation parameter and its variability (velocity and acceleration) and therefore essentially oriented to minimize risk. The integration of EWS and seismogeodetic systems can enhance earthquake preparedness and response significantly. EWS provide rapid alerts, while seismogeodetic systems offer in-depth analysis and data that can improve our understanding of seismic events and their consequences.

In this paper, we present the design and implementation of a Prototype Seismogeodetic System composed of a seismometer combined with a MEMS-type three-component (E, N, Z) accelerometer, a low-cost Tilt Data Logger, a multi-frequency GNSS receiver, and a weather sensor. This prototype has been meticulously designed to exploit the synergies of seismology and geodesy. Our design not only captures the intricate dynamics of seismic waves but also provides precise measurements of ground motion and deformation in near real time. It is located in the Doñana Biological Station, in the Doñana National Park, Huelva, Spain, and this area is characterized by a large complex of faults, giving rise to a complex tectonic evolution and moderate seismic activity as a consequence of the convergence process between the Eurasian and African plates. This prototype is able to generate immediate information to monitor the tectonic activity of the Gulf of Cadiz and adjacent areas, in order to minimize the possible associated risks.

The development and implementation of this prototype system exemplify a significant leap forward in seismogeodetic research. We will explore the design considerations, the hardware and software components, and the methodologies employed to acquire, process, and interpret the data collected by our system.

The main objective of this prototype is to obtain a set of multiparametric data (geodetic, seismic, acelerographic, meteorological, and inclinometry) to provide real-time monitoring and early warning of seismic events that, together with geodetic and geophysical techniques, are able to generate immediate information to monitor the tectonic activity of the Gulf of Cadiz and adjacent areas, in order to minimize the possible associated risks. Other objectives will be the correlation of the different disciplines, time series, and results, in addition to their integration capacity in a regional EWS.

Furthermore, we will present one case study with the results obtained from the 4.4 Mw earthquake that occurred on 1 January 2022 in the Gulf of Cadiz and recorded by the prototype. This case study demonstrates the system’s capabilities and its potential contributions to earthquake science, including early warning systems and post-event analysis. The knowledge and insights gained from this research can significantly enhance our ability to monitor and model seismic events, ultimately leading to more effective earthquake preparedness, mitigation, and risk reduction strategies.

This manuscript is an extension of the proceedings paper: Treatment and Analysis of Multiparametric Time Series from a Seismogeodetic System for Tectonic Monitoring of the Gulf of Cadiz, Spain, published through the 9th International Conference on Time Series and Forecasting (ITISE 2023) [6].

## 2. Design of the Prototype Seismogeodetic System

Real-time seismic wave detection provides valuable information for the rapid location of the epicenter and calculation of the magnitude, playing an important role in early warning systems for earthquakes and tsunamis, as well as in assessing potential associated risks [2,7,8,9]. Recordings of seismic activity are obtained using specific instrumentation (short-period seismometers, broadband seismometers, and/or accelerometers); however, in the case of high-magnitude events, most seismometers can produce saturation in the seismic signal, making it impossible to use these records [1,10].

Although accelerometers do not exhibit saturation, integration of accelerations to velocities or double integration to displacements can cause unreliable drifts due to external anomalies and/or errors in the measurement sensors. This vulnerability of seismic networks is reduced using GNSS networks that produce continuous observations (between 1 Hz and 10 Hz) and which are not affected by such saturation in the signal, providing the value of seismic displacement [11].

Compared to seismic instruments, GNSS networks have a relatively higher noise level, especially in the vertical component [12,13]. The noise level of the cGNSS observations shall be reduced by simultaneously using acceleration measurements recorded at very high frequencies. In smaller earthquakes (less magnitude), seismogeodetic systems complement the information from seismic networks by measuring coseismic, postseismic, or transient displacements, so the combination of both techniques takes advantage of the individual strengths of seismic and geodetic networks while minimizing their weaknesses [1,10,12].

The consequent dangerousness of a high-magnitude earthquake and the possible associated tsunami imply the need and motivation to develop and implement a prototype of a seismogeodetic system that allows the monitoring and surveillance of the tectonic activity in the study area. The proposed prototype comprises geodetic and geophysical techniques capable of providing immediate information on tectonic activity to understand, assess, and minimize potential associated hazards.

The prototype is composed of a multi-frequency GNSS receiver (Leica GR30) that supports all global GNSS constellations and provides position, velocity, and time, a biaxial low-cost MEMS inclinometer (RST DTL202B) with high resolution and sensitivity capable of determining the angle of inclination of a specific point, a low-cost MEMS seismometer/accelerometer (Raspberry Shake RS4D) capable of measuring both strong movements and microseismic activity (velocity and acceleration), and a meteorological sensor (Vaisala WXT520) all in one that measures wind speed and direction, precipitation, barometric pressure, temperature, and relative humidity.

In addition, electrical supply and backup equipment and a communications network for real-time or delayed data transmission have been installed. The RST DTL202B and RS4D sensors were installed in a concrete chamber at a depth of 1 m (Figure 1E,G); the Leica GR30 and Vaisala WXT520 sensors were installed on a metal structure or tripod (Figure 1C,D) near the concrete chamber. The control center located at the LAGC–UCA comprises a network of virtual and physical servers for the acquisition, treatment, processing, backup, and security of data. For more technical specifications of the instrumentation, see Appendix A.

For the transmission and reception of the data produced, a communication network has been established using the VPN protocol, which establishes an encrypted connection over the Internet from a primary host to a destination host and provides connection security and remote control [14]; also, we use RSync service for automatic data storage, synchronization, and replication [15]. For the time synchronization of the inclinometer (DTL202B), we use the NTP service, designed to synchronize the clocks of devices over a network connected to a time server, on a common UTC time base [16]. For the time synchronization of the seismic records (seismometer/accelerometer RS4D), we use a mini GNSS receiver, which connects to different satellites to know their position and navigation time. The data are sent from the Doñana Biological Station to the control center via the CSIC–VPN connection, which offers greater security in the transmission, reception, and availability of the data. The records produced by the prototype are automatically stored on a main NAS server and then distributed to the data processing and filtering modules (Figure 2). To see more technical specifications of the instrumentation, see Appendix A.

### 2.1. Deployed Software Applications

The software used in the development and implementation of the proposed prototype is divided into three modules: Acquisition, Processing, and Data Filtering Modules.

#### 2.1.1. Acquisition Module

This module manages, stores, and visualizes the data produced from the different sensors of the prototype. Seismic data generated by the RS4D seismograph are managed and visualized using the SWARM application (Seismic Wave Analysis and Real-time Monitor), version 3.2.0, developed by USGS Volcano Science Center (VSC), Vancouver, Washington, United States [17], this open source Java application was created to visualize and analyze seismic waveforms in real-time, it can connect to different sources of static data, dynamic data, and common waveforms servers (e.g., Earthworm, Winston, SeisComp, and SeedLink servers). GNSS observations are managed through a local data repository that facilitates data management, sharing, and data searching. The records of the inclinometer DTL202B are managed by the proprietary application DT–Logger–Host, version 1.20.0, developed by RST–Instruments, Kingston St., Canada [18].

#### 2.1.2. Processing Module

This module is dedicated to the treatment, quality control, and processing of the multiparametric data produced by the prototype. The software used for the seismic records is SEISAN, version 12.0, a free multiplatform software useful for processing the waveforms generated by the earthquakes recorded by seismometer/accelerometer RS4D [19]. The data generated by the DTL202B inclinometer are processed with the proprietary DT–Logger–Host software, which also allows us to visualize in real-time the results simply and quickly.

For GNSS data processing, we use the scientific softwares BERNESE, version 5.2, developed by AIUB, and GipsyX, version 1.7, developed by JPL, both of which require a license for use and are under permanent development [20,21]. The GNSS processing techniques used are Precise Point Positioning (PPP), Relative, and Kinematic. In addition, the GR30 GNSS receiver has integrated the VADASE service, which provides real-time velocity and displacement information on movements in the study area produced by the natural or artificial activity.

#### 2.1.3. Filtering Module

In this module, different mathematical and statistical techniques are applied for signal processing, correction of abnormal values, and reduction in the noise level of the time series obtained. For this purpose, data analysis and filtering techniques are used, methodologically grouped into Initial filters (1–σ, 2–σ, and outliers), Analytical filters (Kalman and Wavelets), and statistical filters (ARMA and ARIMA); this filtering software has been developed using the multiplatform and free-to-use statistical language R [22,23,24].

The services and applications are managed from the control center, which has a virtual infrastructure Citrix XenServer Free Edition, version 7.0, developed by Citrix Systems, Inc., Fort Lauderdale, FL, USA, this server has 10 virtual machines with Windows and Linux operating systems. Figure 3 show the hardware and software components used in the development and implementation of the prototype, divided into two groups and three subgroups (Hardware, Communications Components, Virtual Infrastructure, Acquisition, Processing, and Filtering modules).

## 3. Data Management and Processing Techniques

The sensors (seismometer, accelerometer, inclinometer, and GNSS receiver) of the prototype generate raw files in different formats that are treated and processed with specific scientific software (Section 2.1). The RS4D generates daily files at 1 HZ (10 sps) in miniSEED format [25], which is a subset of the SEED standard used for seismic time series data.

The GNSS receiver GR30 provides data every second that are aggregated daily (Raw format), then converted to the standard RINEX format [26] and further processed using different techniques. In addition, this GNSS receiver has the VADASE service, an autonomous solution for real-time velocity and displacement measurement of fast movements produced by natural activities (geological and geophysical phenomena); it is an independent service from the GNSS system, generating velocity and displacement data that are transmitted via NMEA messages and stored in a CSV file. The inclinometer DTL202B produces daily records in 30 s intervals in a CSV file, which are displayed in real time.

### 3.1. GNSS Data Processing

GNSS are widely used to obtain the location of points with an accuracy that depends on the methods and strategies used in the observation. In some cases, it is feasible to reach up to some millimeter repeatability, and to estimate the position of a point, but also its dynamic behavior, i.e., how the position evolves over time, building 3D coordinates time series. To obtain such accuracy, the use of some scientific software is needed. BERNESE 5.2 or GipsyX 1.7 are the most commonly used software packages for these purposes, including different strategies and methods for data analysis. PPP is a method that performs precise position determination in static or kinematic mode using a single GNSS-GPS receiver, if using appropriate antennas and receivers, as well as accurate orbits data and Earth orientation parameters.

NASA’s JPL issued a scientific software package called GipsyX 1.7 to obtain GNSS accurate positioning at millimeter level. GipsyX-PPP is the accurate positioning strategy used to obtain such precise position of a point from GNSS-GPS observations using only one receiver [27]. GipsyX/RTGx is the next generation software package for positioning, navigation, timing, and Earth science. The GipsyX-PPP strategy fixes ambiguities in both static and kinematic modes, reaching repeatability of 2 mm in horizontal and 6.5 mm in vertical for static positioning, which makes it quite useful for geodetic applications [20].

Kinematic processing can be used on a static point, with high frequency sampling data collection, for instance, 1 Hz sampling. It is possible to take advantage of these results to show the motion of a point reached by the seismic waves of an earthquake produced in the location surroundings. In such a way, high-rate GNSS displacements are useful for monitoring long-period ground motion [28]. Point positioning evolution might complement seismometer records, i.e., when the seismic wave intensity produces seismic record saturation, hiding the local behavior. Furthermore, combinations of near-source GNSS and seismic record data provide a broad spectrum of co-seismic motion, including both high frequencies and long periods. Experimental results have shown that high-rate PPP can produce absolute horizontal displacement waveforms at the accuracy of 2–4 mm and absolute vertical displacement waveforms at the sub-centimeter level of accuracy within a short period of time [29].

The GNSS data processing with the BERNESE 5.2 software also uses the PPP technique, which uses GNSS data from a single station and is independent of other reference GNSS stations; this technique reaches its highest accuracy (centimetric) using auxiliary data such as ephemeris, clock corrections, Earth orientation parameters, and atmospheric refraction parameters (ionosphere and troposphere models) [3,27], provided by the International GNSS Services (IGS). One of the particularities of this technique is that it requires resolving satellite ambiguities to ensure accurate coordinates. In addition, the relative positioning method is used, which incorporates simultaneous measurements using different satellites to recognize and cancel orbital errors, satellite clock errors, and signal propagation media (troposphere and ionosphere) through dual satellite–receiver differences. This method allows to calculate the difference between two positions with subcentimetric precision, so it requires one of the GNSS stations to be recognized through a reference frame.

The best accuracies are obtained with the relative processing technique, which focuses on calculating the distances between the GNSS antenna and the satellite through the carrier wave itself using interferometric processes. The final calculation is obtained by combining this method with the differential method; i.e., one of the receivers must be at a point with known and reliable coordinates [22]. The records obtained from the GNSS receiver form time series initially in geocentric Cartesian coordinates (X, Y, Z), to facilitate the concept of displacement on Earth’s surface; these coordinates are transformed into a topocentric system with east, north, and vertical components. The analysis of these series provides the displacement vector, as well as the anomalies that have occurred in the time period defined by the series [30].

As for the geodetic control, the variations in the absolute coordinates obtained through a process of calculation and geodetic adjustment in both Cartesian coordinates (X, Y, Z) and topocentric coordinates (E, N, U) are analyzed, using different techniques that guarantee millimeter accuracy to provide the most optimal results of the tectonic surface deformation of the study area; this involves the use of precise auxiliary parameters such as accurate ephemerides, corrections of satellite clocks, and tropospheric models, as well as data processing and filtering methods capable of jointly managing the results [23,30].

In the analysis phase, different mathematical and statistical techniques for data filtering (1–σ, 2–σ, Kalman, and Wavelets) are used to detect the behavior of linear and non-linear time series. Initially, outliers are filtered out, which can be introduced by the physical environment or by the instrument itself. The use of these filtering techniques applied to each time series minimizes noise and improves the resolution or accuracy of the results, more specifically in the determination of the velocity or displacement (E, N, U) of each point. The commonly used filtering techniques are 1–σ and 2–σ, which search for outliers in the series, considering their deviations through a linear regression method, and then a predictive–corrective Kalman filter is applied and in turn a harmonic value analysis is performed using the Wavelets filtering technique to reduce the noise of the time series.

In Figure 4 and Figure 5, we show the time series analysis of cGNSS DONA station, which is a fundamental part of the prototype seismogeodetic system. The relative processing technique was used with the cGNSS reference stations VILL (Villafranca) and YEBE (Yebes) located in the province of Madrid, Spain; both stations belong to the international network IGS. GNSS processing was performed with the software BERNESE 5.2, using the ITRF14 reference frame; the years analyzed were from 2016 to 2022. The original time series show the deformation values: East = 23.2 mm/y, North = 15.1 mm/y, Up = −9.3 mm/y. Using the CATS filter, the deformation values are East = 22.8 mm/y, North = 15.0 mm/y, Up = −9.1 mm/y, (Figure 4).

### 3.2. Seismic Data Processing

The seismic signals recorded by the RS4D seismometer/accelerometer are visualized using the software SEISAN 12.0, which has several subroutines and services used for analog and digital analysis of seismic recordings. The SEISAN software can locate a seismic event, determine the spectral parameters and azimuth of the arrival signal, and visualize and determine the epicenters and depth of the seismic event [31]. In Figure 6, a seismic signal is observed in real-time, the EHZ component corresponds to the vertical component of the seismometer, and the ENE, ENN, and ENZ components correspond to the accelerometer. In this image, a simple simulation of a seismic event is displayed in order to know the resolution of the sensor and check the quality of the generated data. The data acquisition system stores records in miniSEED format in a user-defined directory structure; these data are converted to a new multi-channel format containing three recording channels for the accelerometer and one recording channel for the seismograph in its vertical component. Later, these channels are stored in a database; these records consist of very small files (ASCII format), containing the information of the P (Primary) and S (Secondary) phases of an event, in addition to important parameters such as arrival times of the seismic wave, duration, amplitude, periodicity, azimuth, and apparent velocity.

The signal identification process begins with the selection of events based on a criterion (e.g., signal-to-noise ratio). Once the noise level of the seismogram is identified, the next step is to identify all other disturbances relative to this noise signal. Then, the time of the trace and its amplitude are delineated, and the classification is based on the duration of the signal, its type, and its more or less monochromatic character, and the energy content of the frequency bands where the signal originates. For this purpose, frequency spectrograms are used to represent the spectrum of the signal as a function of time, which allows a comparison of the waveforms based on the accumulated energy.

The seismic data processing consists of individual analysis of the recordings, amplification of each trace, and application of filtering techniques and noise analysis. Once a change in signal and noise level is detected, the waveforms are reduced to measure the arrival, start, and end times of the event and calculate its magnitude, amplitude, epicentral location, and the time difference between the arrival of the waves P and S. The filtering techniques are intended to reduce the noise level of the seismic signal, and also improve the visualization of the waveforms and their characteristics, (Figure 7).

## 4. Application of the Designed Prototype to the Case Study: Earthquake 4.4 Mw, 1 January 2022, Gulf of Cadiz

### 4.1. Seismotectonic Settings of the Gulf of Cadiz

The south of the Iberian Península and North Africa region corresponds to the transition between the oceanic edge and the continental edge where the Iberian Peninsula and Africa meet in the direction of Tunisia, includes the Betic mountain ranges, the Gulf of Cadiz, the Alboran Sea, and the northern part of Morocco, characterized by a large complex of faults, giving rise to a complex tectonic evolution and moderate seismic activity as a consequence of the convergence process between the Eurasian and African plates. In the Alboran Sea region, north of Morocco and the southeastern peninsula, seismicity is diffuse, which makes it difficult to identify the plate boundary. Most of the seismic activity is concentrated in the shear zones of the Eastern Betic Shear Zone (EBSZ) and its prolongation in the Alboran Sea along the Trans-Alboran Shear Zone (TASZ) [32,33,34,35,36] (Figure 8).

In the Gulf of Cadiz, the seismic activity is distributed in the E–W direction along a 100 km wide band located north of the Gulf, this tectonic activity (according to the magnitude, intensity, location, depth, and other characteristics of the event) leads to the possibility of tsunami occurrence in the area [37]. The tsunami that has produced the greatest natural catastrophe in Spain was recorded on 1 November 1755, as a result of an earthquake of magnitude 8.5 Mw, located about 200 km from the cape of San Vicente in the direction of S–W [38,39].

Figure 8 show the hypocenters distribution of >3.5 Mw seismic events recorded in the study area from 2015 to 2022. Dense seismic activity is observed in four defined clusters; from left to right, we find (A) the Gorringe Bank, (B) São Vicente Canyon area, (C) the SW terminus of the Horseshoe fault, and (D) in the area of the Guadalquivir Bank and the Gulf of Cadiz accumulation wedge. These seismic clusters are elongate and consistent; these parallel the active faults in the region NNE–SSW and NE–SW, for São Vicente Canyon and Gorringe Bank, respectively. However, for the Horseshoe seismic cluster, the movement is from WNW–ESE to NW–SE, slightly rotated clockwise with respect to the faults of SW Iberian margin. In addition, for the Guadalquivir bank, the seismic clusters parallel the faults on the Iberian continental margin SW [40,41].

The main tsunami-generating faults and submarine landslides in the Gulf of Cadiz are the overthrust systems of the Gorringe Ridge, the Marquês de Pombal, the São Vicente Canyon, and the high-susceptibility horseshoe faults; these are the areas of greatest potential vulnerability to submarine landslides. In addition, this area is characterized by the occurrence of large earthquakes in long intervals, which have the potential to trigger tsunamis (Figure 8) [42,43].

### 4.2. Results

In the last two years, several earthquakes have been recorded in the Gulf of Cadiz; however, they have not been of high magnitude nor have they occurred very close to the seismogeodetic system installed in the Doñana Biological Station. However, a case study has been included to illustrate the purpose and scope of the prototype. The earthquake analyzed in this manuscript was the one that occurred at 21:03:49 (UTC) on 1 January 2022, of magnitude 4.4 MW and depth 6 km, whose epicenter was located about 130 km southwest of Doñana, Huelva, Spain. This event represents the common type of seismicity in the Gulf of Cadiz characterized by being a constant activity of moderate intensity at shallow depths (<40 km) and with hypocenters (often) in the vicinity of the banks of the Guadalquivir River, (Figure 9).

The analysis of the seismic signal from this earthquake shows an acceptable signal-to-noise ratio, as shown in Figure 7B; where the unfiltered event is shown. To improve the observation of the signal, a bandpass filter of 2–15 Hz was applied, in which the phases and the duration of the event (290 s) can be clearly seen. To view the characteristics of the phases, Figure 7C; was amplified, where the arrival of the impulsive P-wave and the arrival of the S-wave can be seen, showing a S and P waves of 14 s. Furthermore, the seismic signal has a broadband spectrogram that can be seen in Figure 10B.

The study of the focal mechanism shows the parameters double pair, plane A: average azimuth of 112°, average dip of 89°, and a slip angle of 156°. Double pair, plane B: average azimuth of 203°, average dip of 86°, and slip angle of 1°. This solution presents strike-slip faulting with NW–SE trending P axis, according to the NW–SE to WNW–ESE direction of Eurasian plate convergence. This mechanism is similar to previous moment tensor solutions in the Gulf of Cadiz [44,45,46] (Figure 9).

This seismic event is located within the elongate clusters of the Guadalquivir Bank and parallel to the fault according to NW–SE, possibly located in the transition zone between the continental and oceanic crust, which is about 100 km south of the coastline and 60 km wide. This area has been described as one where there is an abrupt lateral change in crustal thickness (by 20 km) between the continental margin and oceanic crust in the middle of the Gulf of Cadiz, where the strongest transition occurs at 160–170 km [47].

Regarding the kinematic GNSS processing, we observe that the 3D evolution of the GNSS receiver antenna position occurred 45 s after the seismic event; there is also a small but significant displacement in the N and U components; however, component E shows a smaller displacement than the previous ones (Figure 11). This earthquake lacks features that allow for the production of highly significant GNSS kinematic records, to be correlated with seismic, accelerometric, or inclinometry records. Figure 12 shows the inclinometry results of the analyzed seismic event.

On the other hand, the Andalusian Institute of Geophysics of the University of Granada has installed a network of accelerometric equipment at various strategic points near the coast of the Gulf of Cadiz; this network helps to verify the results generated by the sensors of the prototype. Below, we show the accelerographic signal from the ALMT and LEPE sensors located in the municipalities of Almonte and Lepe in the province of Huelva, close to the Doñana Biological Station; these signals correspond to the 5.4 Mw seismic event that occurred on 14 August 2022 in the Gulf of Cadiz (Figure 13 and Figure 14).

## 5. Conclusions

The seismogeodetic prototype presented has demonstrated the ability to detect earthquakes of at least 4.4 Mw, such as the one registered in the Gulf of Cádiz on 1 January 2022. Earthquakes of greater magnitude should be easily recorded since the sensors that make up said prototype are directly correlational (GNSS receiver, inclinometer, and seismometer/accelerometer), guaranteeing the availability of data and results in real time. The presence of accelerometers and GNSS receivers provides valuable records of an event as seismic sensors become saturated in large earthquakes. All this allows us to ensure that the proposed prototype would be very suitable for integration into a regional early warning system due to its reliability, simplicity, and low cost since the equipment is based on minicomputers that incorporate MEMS technologies. The use of modular equipment that allows for easy maintenance and replacement with devices that can be found on the market has an impact on the fact that this prototype can be deployed in regions that do not have large resources, increasing knowledge of the tectonic behavior of the area, and reducing possible risks associated with dangerous geological events.

The seismogeodetic prototype presented has demonstrated its effectiveness in detecting earthquakes with a magnitude of at least 4.4 Mw. The system’s capabilities extend beyond this threshold, so it can record even more significant seismic events. This is primarily due to the direct correlation between the prototype’s sensors, which include a GNSS receiver, an inclinometer, and a seismometer/accelerometer. This correlation ensures the continuous availability of data and results in real time.

Notably, the inclusion of accelerometers and GNSS receivers adds a layer of resilience to the system as these components continue to function even when seismic sensors become saturated during large earthquakes; this robustness makes the proposed prototype highly suitable for integration into a regional early warning system. Its reliability, simplicity, and cost-effectiveness, based on minicomputers incorporating MEMS technologies, make it an attractive choice for regions with limited resources. In summary, the seismogeodetic prototype presented in this study holds promise for contributing to earthquake monitoring and early warning efforts in regions with diverse resource constraints.

While the prototype has demonstrated its ability to detect earthquakes of moderate magnitude, it would be valuable to extend its study to large-magnitude earthquakes. This includes examining how the system responds to extreme ground motion, saturation of sensors, and the impact on the quality of recorded data. Such research can contribute to the development of advanced algorithms for handling large-magnitude seismic events and further refine the system’s early warning capabilities.

In the case study presented (4.4 Mw earthquake that occurred on 1 January 2022, located in the Gulf of Cádiz, recorded by the prototype), we can observe that the results show that there is a small but significant displacement in components N and U; however, in component E, it shows a smaller displacement than the previous ones. This earthquake lacks characteristics that allow the production of highly significant GNSS kinematic records, which can be correlated with seismic records.

One potential avenue for future research involves a more in-depth analysis of data filtering methods (Kalman, Wavelets, ARMA, and ARIMA) generated by the prototype. Investigating the response of various filtration techniques to different seismic events can provide insights into the best practices for data preprocessing and filtering. This comparative analysis can lead to the refinement of data processing algorithms, enabling more accurate and robust event detection and characterization.

## Figures and Tables

**Figure 1 sensors-23-08986-f001:**
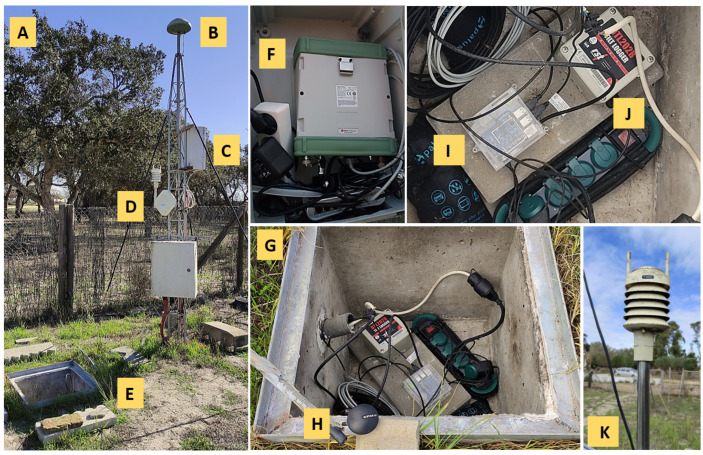
Instrumentation located in the Doñana National Park, Huelva, Spain: (**A**) the prototype seismogeodetic system, (**B**) metal tripod that supports the Leica Geodetic (AR10) antenna, Leica GNSS receiver, and weather sensor, (**C**) box containing the GNSS receiver and main communication switch, (**D**) Vaisala weather sensor (WXT520), (**E**) concrete chamber located 1m away from the tripod, (**F**) Leica GR30 GNSS receiver, (**G**) contents of the concrete chamber: Seismometer, Inclinometer, communications connectors, power supply connectors, and desiccant bags that prevent humidity, (**H**) GNSS receiver UBX–M8030, (**I**) Raspberry Shake 4D Seismometer/Accelerometer, (**J**) Biaxial Digital Tilt Logger DTL202B, and (**K**) Vaisala weather sensor, owned by AEMET.

**Figure 2 sensors-23-08986-f002:**
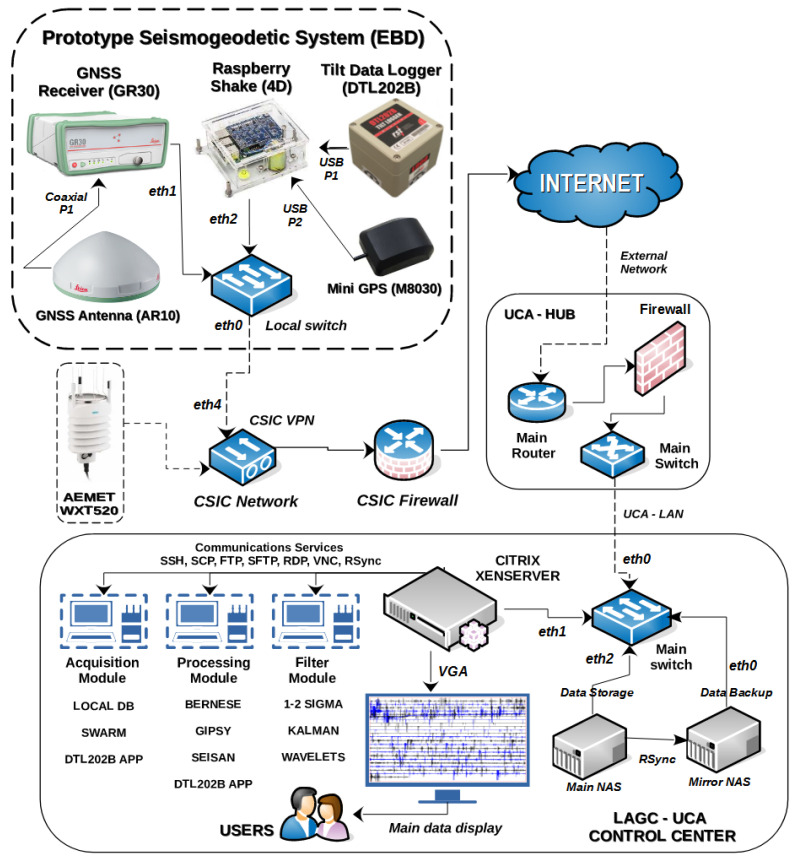
Network diagram and hardware components of the prototype seismogeodetic system (communications, sensors, servers, virtual machines, NAS, mirror backup, etc.); it is divided into three parts: Prototype Seismogeodetic (Doñana Station), UCA–HUB, and Control Center (LAGC). Initially, the prototype, and the UCA–HUB are interconnected by the VPN service provided by the CSIC, facilitating data transmission over the Internet to the management and control center, which has a Citrix XenServer virtual infrastructure with virtual machines that have services and applications dedicated to the automatic acquisition, processing, visualization, and filtering of data produced.

**Figure 3 sensors-23-08986-f003:**
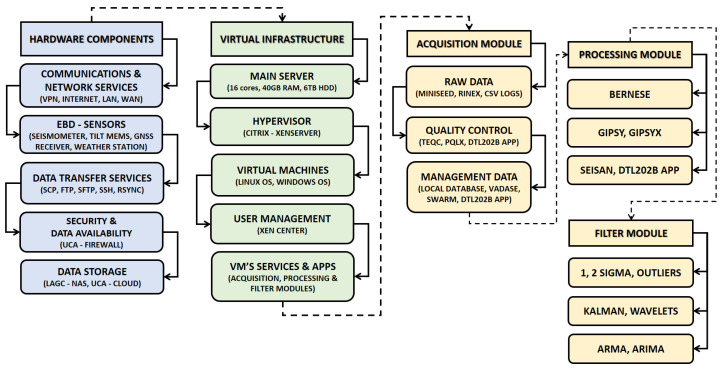
Structure of the components involved in the implementation of the prototype; it is divided into two groups and three subgroups that show the components of hardware, services, communications, and virtual infrastructure of the control center (LAGC–UCA), and Doñana Station, as well as the virtual machines that contain the acquisition, processing, and filtering modules.

**Figure 4 sensors-23-08986-f004:**
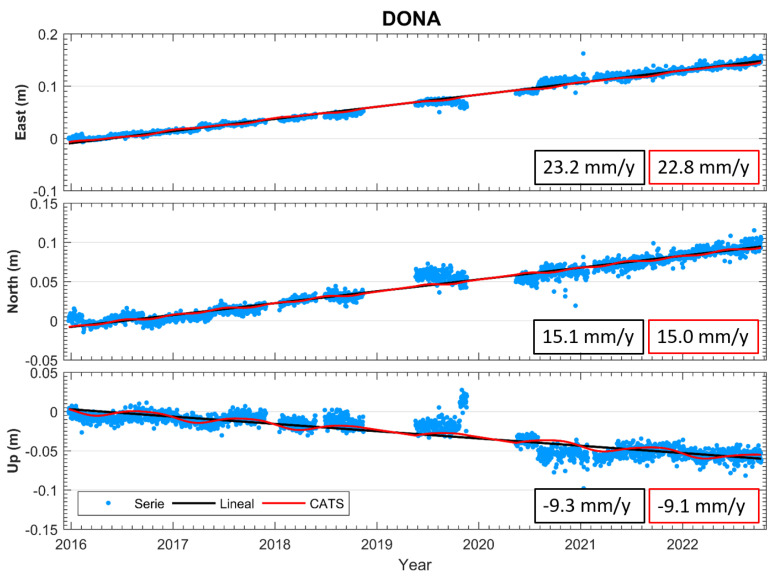
Results (E, N, U) of the DONA station time series; the GNSS processing was performed with the BERNESE 5.2 software using ITRF14. This figure shows the time series with the linear fit and the CATS filter, as well as the velocities per component.

**Figure 5 sensors-23-08986-f005:**
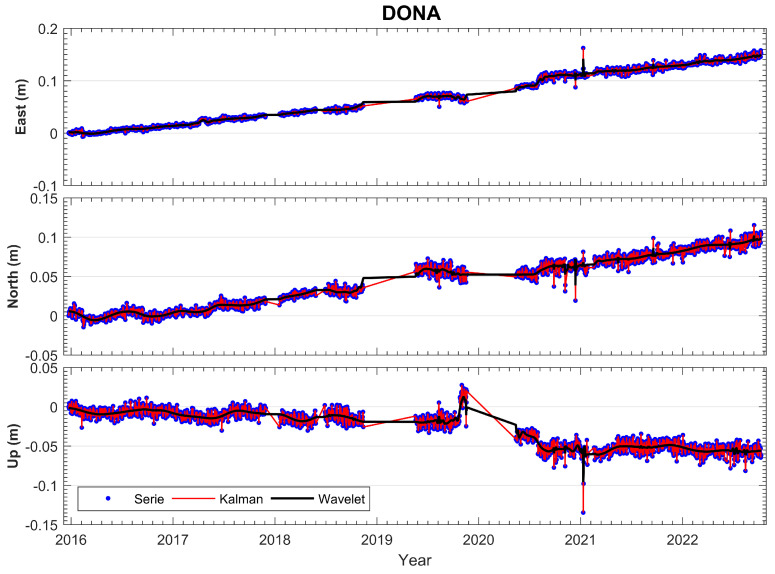
Results (E, N, U) of the DONA station time series; the GNSS processing was performed with the BERNESE 5.2 software using ITRF14. In addition, Kalman and Wavelets filters were applied.

**Figure 6 sensors-23-08986-f006:**
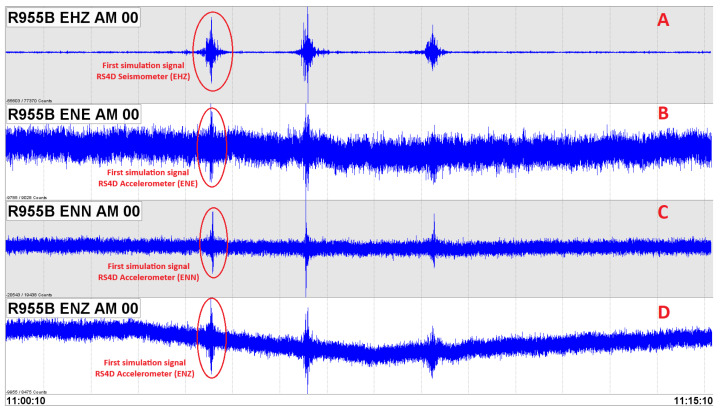
Seismogram with a simulation of a seismic event (EHZ, ENE, ENN, and ENZ components) to know the resolution of Raspberry Shake RS4D seismometer/accelerometer and check the quality of the generated data.

**Figure 7 sensors-23-08986-f007:**
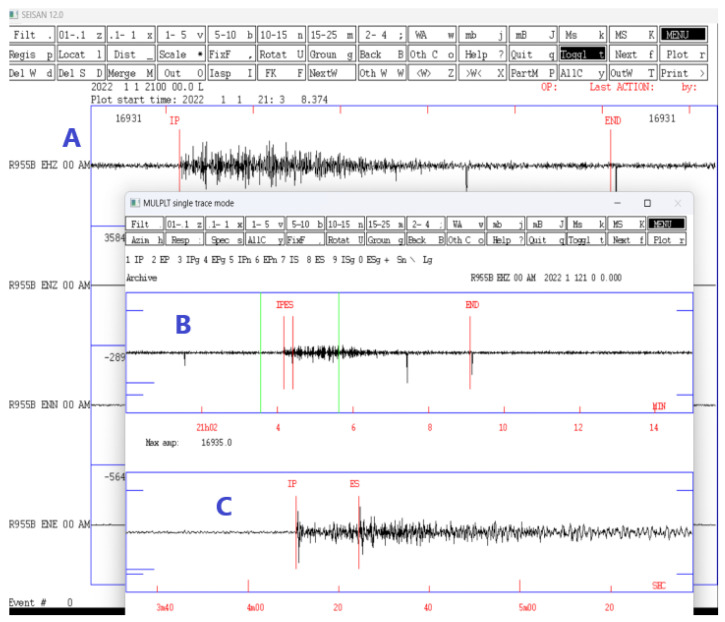
Seismic events display on SEISAN 12.0 software for seismic analysis. (**A**) Seismogram of the earthquake that occurred at 22:03:49 on 1 January 2022, recorded on Z channel of the RS4D seismometer with a filter of 2–15 Hz. (**B**) Unfiltered seismogram of the same event with the waves phases P and S, and coda selected. (**C**) Signal amplification, and impulsive arrival of the P-wave and arrival of the S-wave.

**Figure 8 sensors-23-08986-f008:**
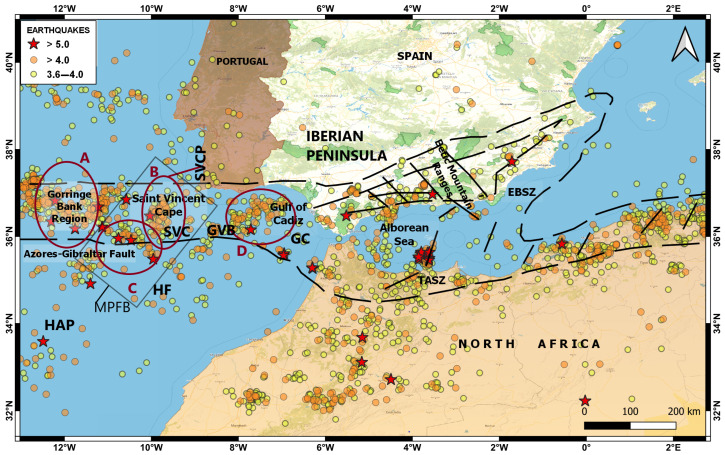
Map showing the geodynamic context, seismic activity (2015–2022), and main faults of the southern region of the Iberian Peninsula and North Africa. The most important faults are Gorringe Bank Region, Gulf of Cadiz (GC), Azores–Gibraltar Fault, Saint Vincent Cape (SVCP), Alboran Sea, Betic Mountain Ranges, Eastern Betic Shear Zone (EBSZ), Trans-Alboran Shear Zone (TASZ), Horseshoe Abyssal Plain (HAP), Horseshoe fault (HF), São Vicente Canyon (SVC), Guadalquivir Bank (GVB), and Marquês de Pombal Fault Block (MPFB). In addition, clusters A, B, C, and D are shown, which reflect the concentration and distribution of the seismic epicenters.

**Figure 9 sensors-23-08986-f009:**
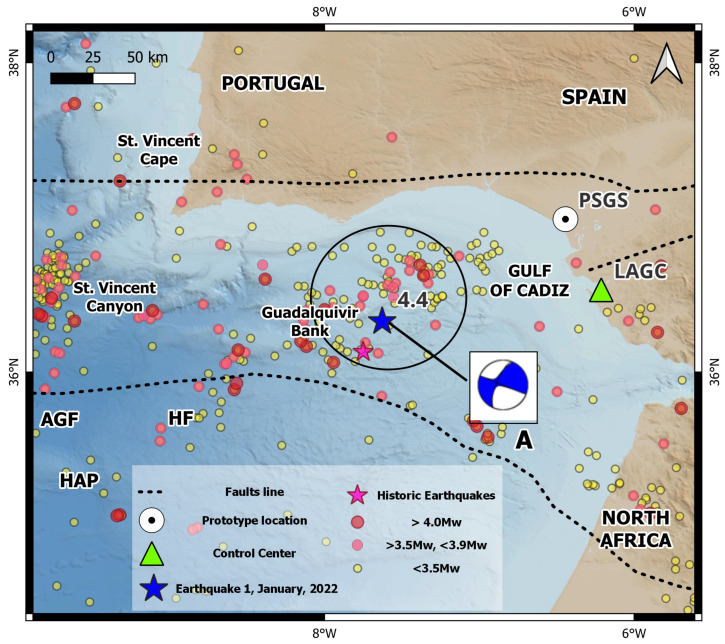
Map showing the location of the 4.4 Mw earthquake that occurred on 1 January 2022 in the Gulf of Cadiz (Lat: 36.3276, Lon: −7.6271, depth: 6 km) recorded by the RS4D seismometer/accelerometer. We also show seismic events of different magnitudes that occurred between 2005 and 2022 in the Gulf of Cadiz and adjacent areas (data taken from the public seismic catalog of IGN), the location of the Seismogeodesic System in the Doñana Biological Station, Huelva, Spain, the Control Center (LAGC–UCA), and the focal mechanism produced by the studied earthquake (A).

**Figure 10 sensors-23-08986-f010:**
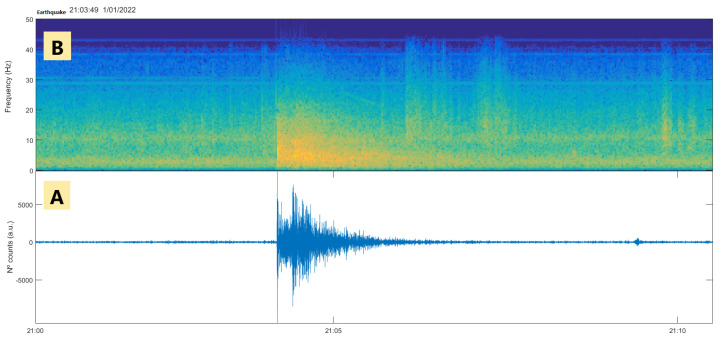
Seismogram (**A**) and spectrogram (**B**) of the earthquake that occurred on 1 January 2022 at 22:03:49 registered by the RS4D seismometer/accelerometer integrated in the prototype. In this seismic signal, a low signal-to-noise ratio was found in certain periods of time, which allowed the use of a first filter of 0.5 Hz to 10 Hz and a later one of 2 Hz to 8 Hz.

**Figure 11 sensors-23-08986-f011:**
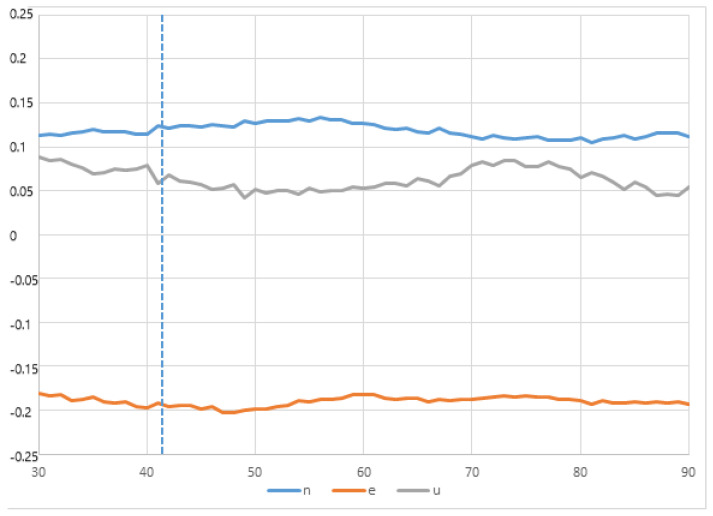
East, North, and Vertical components of the GNSS-GPS time series (1 Hz sample rate) for the position of the GR30 GNSS receiver seconds after the magnitude 4.4 Mw earthquake of 1 January 2022, with epicenter about 130 km southwest of Doñana, Huelva, Spain. A small change in the trend is shown 45 s (approximately) after the event occurred; this corresponds to the arrival of the seismic wave.

**Figure 12 sensors-23-08986-f012:**
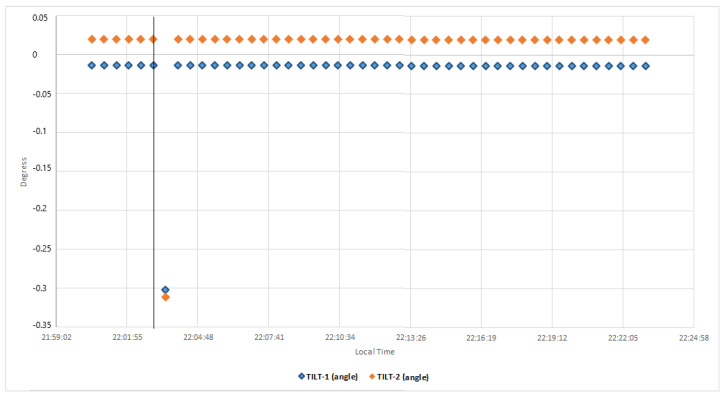
Inclinometry results (30 s sample rate) where the displacement produced in both sensors (Tilt 1, Tilt2) is observed and that corresponds to the arrival of the seismic wave of the 4.4 Mw earthquake that occurred on 1 January 2022 in the gulf of Cadiz.

**Figure 13 sensors-23-08986-f013:**
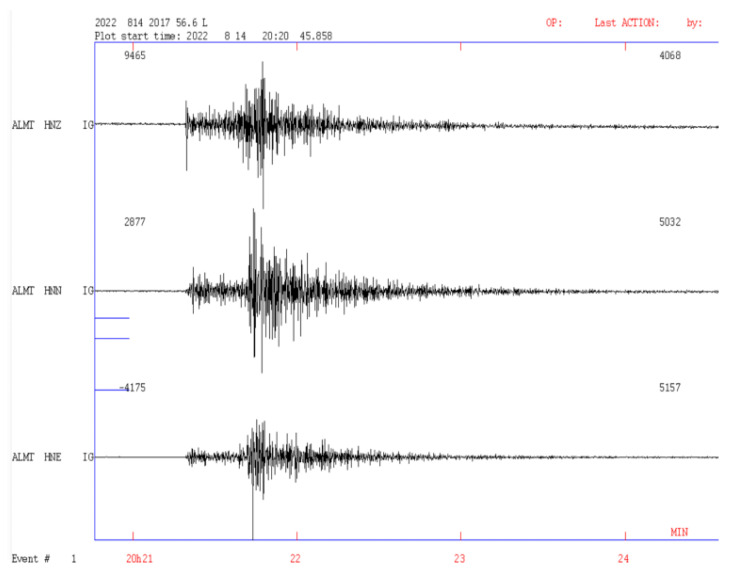
Accelerographic signals from the ALMT (Almonte) station corresponding to the 5.4 Mw earthquake that occurred on 14 August 2022 in the Gulf of Cadiz.

**Figure 14 sensors-23-08986-f014:**
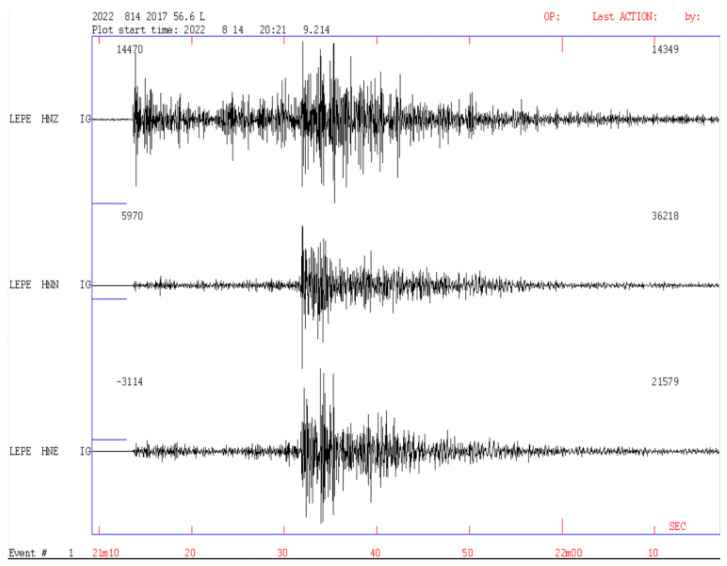
Accelerographic signals from the LEPE station corresponding to the 5.4 Mw earthquake that occurred on 14 August 2022 in the Gulf of Cadiz.

## Data Availability

The seismic data shown in this manuscript belong to the public seismic catalog of the National Geographic Institute of Spain (IGN), https://www.ign.es/web/ign/portal/sis-catalogo-terremotos (accessed on 4 November 2023). The data generated by the seismogeodetic system belong to the laboratory of astronomy, geodesy, and cartography of the University of Cádiz (LAGC-UCA), https://lagc.uca.es/ (accessed on 4 November 2023).

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
