# Peer review of "Design and Implementation of a Prototype Seismogeodetic System for Tectonic Monitoringâ€"

_sensors, 2023, doi:10.3390/s23218986_

Round 1
Reviewer 1 Report
Comments and Suggestions for Authors
The paper describes a seismogeodetic and tectonic monitoring installed in the Iberian Peninsula.
A large portion of the manuscript is just a description of the hardware and software use for monitoring seismic activity. The description does not add any novelty and reads like information from a data-sheet or the manufacturer website. There is a good amount of text regarding internet protocols, etc. which I found do not add much to the purpose of the paper. In my opinion, all this material could be "Appendix material".
The section on data processing lacks details. Again it consists of a very high-level description of the signal processing techniques. Furthermore, they appear to be standard techniques such as filtering, and outliers detection.
The last section of the paper does not seem to provide novelty regarding seismic activity detection.
The introduction of the paper claims the system will aid measuring seismic activity for high-magnitude earthquakes. However, little is said regarding this issue throughout the paper.
The authors should reconsider the scope and purpose of the manuscript and describe the novelty of their implementation rather than simply describing their elements. Is the novelty in the way the sensors are combined? Is there any new signal processing method (e.g. sensor fusion) used? etc.
English is fine in my opinion.
Author Response
Thank you very much for the suggestions, attached the report.

Reviewer 2 Report
Comments and Suggestions for Authors
Dear Authors,
This paper implements a prototype system based on seismogeodetic techniques to detect and monitor the tectonic activity in the Gulf of Cadiz region and adjacent areas in which important seismic events occur produced by the interaction of the Eurasian and African plates. The paper is well structured, and i only have a few suggestions for details.
All the best.
1. Introduction should be clearly presented to highlight main ideas and motivation behind the proposed research. Please include and clearly state research question and contributions of proposed study in Introduction. Moreover, research contributions are not clearly emphasized, please clarify them.
2. More methods should be included in comparative analysis to validate proposed approach..
3. Conclusion should be extended to include more details regarding the future work and limitations of proposed study.
4. Some references are missing parts, such as pp., publisher, etc.
Comments on the Quality of English Language
Minor editing of English language required
Author Response

(The authors gave the same response as above.)

Round 2
Reviewer 1 Report
Comments and Suggestions for Authors
Authors have addressed my concerns. I think the paper is ready for publication.
Author Response
Thank you very much for your excellent recommendations that have significantly improved the structure and sections of the article.